# The Diagnostic Utility of Prenatal Microarray in High-Risk Pregnancies: A Single-Center Experience in Enhancing Reproductive Care and Risk Stratification

**DOI:** 10.3390/diagnostics15172129

**Published:** 2025-08-23

**Authors:** Abdullatif Bakır, Mustafa Tarık Alay, Umut Can Tekbaş, Sadun Sucu, İrem Kalay, Hanife Saat

**Affiliations:** 1Department of Medical Genetics, Ankara Etlik City Hospital, Ankara 06170, Turkey; mtarikalay@gmail.com (M.T.A.); umutcantekbas@gmail.com (U.C.T.); hanifesaat@gmail.com (H.S.); 2Department of Obstetrics and Gynecology, Ankara Etlik City Hospital, Ankara 06170, Turkey; medical.academic.sucu@gmail.com; 3Department of Medical Genetics, Umraniye Training and Research Hospital, İstanbul 34764, Turkey; dr.iremgenetic@gmail.com

**Keywords:** microarray analysis, prenatal diagnosis, amniocentesis, chorionic villus sampling, fetal blood sampling

## Abstract

**Background/Objective:** Prenatal cytogenetic testing is essential for pregnant women who are at high risk of having a child with a chromosomal abnormality. While conventional karyotyping detects large aneuploidies and structural rearrangements (>5–10 Mb), chromosomal microarray analysis (CMA) identifies smaller copy number variants (CNVs), increasing the diagnostic yield by approximately 5%. CMA is now recommended as the first-line test for evaluating fetal structural anomalies that are detected by ultrasound. **Method:** From March 2023 to September 2024, we analyzed 344 prenatal samples using conventional karyotyping and SNP-based CMA. Karyotyping was performed via flask culture, and CMA was conducted using the Infinium Global Screening Array Cyto (GSA-Cyto) on the Illumina iScan platform. We interpreted the CNVs using NxClinical v6.0 and curated databases including ClinVar, DECIPHER, OMIM, and ClinGen, among others. Our results aligned with the GRCh37/hg19 reference genome. **Results:** Chromosomal abnormalities were identified in 57/344 cases (16.5%). Of these, 39 cases were numerical chromosomal anomalies, and 18 cases were pathogenic or likely pathogenic CNVs. Notably, 11 CNVs (3.2%) were undetectable by conventional karyotyping, emphasizing the added value of CMA. **Conclusions:** CMA enhances the prenatal diagnostic accuracy by detecting submicroscopic CNVs that are not visible with conventional methods, supporting the routine use of this analysis in prenatal genetic evaluation.

## 1. Introduction

During prenatal diagnosis, conventional cytogenetic analyses are typically the first option offered to women at high risk of having a child with a chromosomal abnormality. Conventional cytogenetic analysis has historically served as the mainstay in prenatal diagnosis for couples at increased risk of having a fetus with chromosomal abnormalities. These techniques allow for the detection of aneuploidies and large chromosomal rearrangements, generally ranging from 5 to 10 megabases (Mb) in size. In recent years, developments in prenatal genetic testing have facilitated the use of additional approaches, such as fluorescence in situ hybridization (FISH) and chromosomal microarray analysis (CMA), alongside traditional methods. CMA, which includes comparative genomic hybridization arrays and single-nucleotide polymorphism (SNP) arrays, broadens the diagnostic capability depending on the design of the platform and the analytical strategy applied. CMA is a molecular cytogenetic approach that enables the identification of microscopic and submicroscopic chromosomal abnormalities smaller than 5 Mb with high sensitivity. Specifically, SNP-based CMA can detect alterations as small as 50–100 kilobases (kb). With a resolution approximately 100 times greater than that of standard karyotyping, CMA has made it possible to uncover numerous CNVs, including both benign polymorphic variants and novel pathogenic changes [1]. Over the past decade, its use in prenatal testing has substantially improved the accuracy and timing of genetic diagnoses [2]. Clinically significant results are observed in around 5% of cases (range: 2.3–8.3%) [3,4]. Nevertheless, CMA cannot detect low-level mosaicism or balanced chromosomal rearrangements, and it frequently reveals variants of uncertain significance (VOUS), which can complicate interpretation. The increasing adoption of genome-wide, high-resolution platforms has contributed to a growing rate of VOUS detection.

A high-risk pregnancy may be indicated by factors such as risk determination in prenatal maternal serum screening (MSS) tests, advanced maternal age (AMA), intrauterine growth restriction (IUGR), increased nuchal translucency (NT), the detection of a structural anomaly in a prenatal US, a family history of chromosomal abnormalities, and risk determination in non-invasive prenatal tests (NIPT). CMA has become the first-line technique to perform during genetic follow-up when structural anomalies are detected in prenatal ultrasound [5].

The rate of pathogenic CNV detection in the prenatal setting varies by indication; for instance, studies have reported detection rates from 0% to 15% in fetuses with increased NT (≥2.5–3.5 mm, corresponding to the 95th–99th percentile) or cystic hygroma, and up to 18–22% in cases with congenital heart disease (CHD) [6,7]. The most common genetic etiologies in CHD include trisomy 21, trisomy 18, and 22q11 microdeletion [8,9]. Other systems frequently affected in CMA findings include the skeletal, genitourinary, and central nervous systems (CNS) [10,11,12]. Evidence remains limited on the frequency of clinically relevant CNVs in fetuses showing ultrasound soft markers such as echogenic intracardiac focus, mild ventriculomegaly, enlarged cisterna magna, choroid plexus cysts, increased NT, echogenic bowel, or mild hydronephrosis. Similarly, the prevalence of such variants in pregnancies referred for invasive testing due to AMA, abnormal MSS, or abnormal NIPT results is not well established. In this study, we outline a retrospective cohort study on high-risk pregnancies at our center, in which both G-banded karyotyping and CMA were carried out. We performed these analyses because a structural anomaly was observed during the prenatal diagnosis, or we were suspicious of a high-risk pregnancy.

## 2. Materials and Methods

### 2.1. Ethics

This research received ethical approval from the Ankara Etlik City Hospital Scientific Research Evaluation and Ethics Committee, with the following document number: AESH-BADEK-2024-876.

In this study, we evaluated the CMA results of high-risk pregnancies that underwent prenatal diagnosis at Ankara Etlik City Hospital between March 2023 and September 2024. At the time of writing, 344 pregnant women have undergone CMA as part of their prenatal diagnosis, and various chromosomal anomalies have been detected in 57 patients. The data used in this research only includes genetic report results, and access will be restricted to researchers. The identities of the patients included in this study will remain confidential.

### 2.2. Inclusion Criteria

Patients with an indication for prenatal CMA due to risks in the prenatal MSS; increased NT, AMA, IUGR; increased NT; the detection of a structural anomaly or a soft marker in a prenatal US; a family history of chromosomal abnormalities; and risk determination in NIPTs.Patients who signed an informed consent form and agreed to undergo prenatal genetic testing.

### 2.3. Exclusion Criteria

Patients within whom biochemical risks were identified during prenatal screening but who did not have increased NT, prenatal USG abnormalities, or parental karyotype anomalies, which are required for a prenatal CMA indication.Patients who did not sign the informed consent form and declined to undergo prenatal genetic testing.

### 2.4. Parameters to Be Examined

Presence of possible aneuploidy;Presence of possible microdeletions/microduplications;Mosaicism;Uniparental disomy (UPD).

### 2.5. Karyotype Analysis

Samples obtained from the patients, such as chorionic villus sampling (CVS), amniocentesis (AS), or fetal cordocentesis, depending on their gestational age, were subjected to cell culture using the flask method for genetic testing. The metaphases obtained after culture were analyzed at the 450–550 GTG band level using the CytoVision^®^ platform.

### 2.6. CMA

In patients who met the inclusion criteria, we performed CMA in addition to conventional cytogenetic analysis. For this purpose, DNA was first isolated from the prenatal samples. Maternal DNA was also isolated by obtaining a peripheral blood sample from the mother. The isolated DNAs were compared to exclude maternal contamination, and then CMA was initiated. Contaminated materials were not included in the study. A new sample was requested from patients in whom contamination was detected, and repeat testing was planned for those who agreed to provide a new sample. CMA was performed using Infinium Global Screening Array Cyto (GSA-Cyto) chips on the Illumina iScan platform. Copy number variations were detected and visualized using the NxClinical (v.6.0) analysis software developed by Biodiscovery. The relevant genomic positions were reported based on the Human Genome Build 37 (GRCh37/hg19) reference assembly. The obtained data were evaluated using currently available databases, including PubMed, OMIM, DGV, ClinVar, DECIPHER, and ClinGen. Variant interpretation was performed, and only pathogenic and likely pathogenic variants were reported. For patients in whom a VOUS was detected, parental testing was not planned in order to avoid prolonging the reporting process.

### 2.7. Statistical Analysis

Statistical analysis of the data in this study was performed using IBM SPSS Statistics for Windows, Version 30.0 (IBM Corp., Armonk, NY, USA) or Windows and the R programming language. Quantitative variables (discrete or continuous numerical variables) were expressed as mean and standard deviation when they showed normal distribution; otherwise, they were expressed as median and interquartile range (IQR). Qualitative (nominal and ordinal) variables were explained using numbers and percentages. Ordinal variables were arranged in the table according to their hierarchical order.

## 3. Results

The pregnant women enrolled in this study were aged between 17 and 45 years, with an average age of 30.66 years, and they were 9–33 weeks pregnant, with an average of ~19.93 ± 1.63 weeks. A total of 344 prenatal samples were analyzed using CMA. The clinical indications for testing included abnormal ultrasound findings, congenital anomalies, multiple anomalies, increased NT, CNS anomalies, skeletal anomalies, biochemical risk, family history, hydrops fetalis, positive NIPT results, AMA, cystic hygroma, IUGR, and amniotic fluid abnormalities (anhydramnios/oligohydramnios). The distribution of abnormal findings according to clinical indications is summarized in Table 1.

Overall, the combination of karyotyping and CMA detected chromosomal abnormalities in 57 cases, corresponding to a total abnormality detection rate of 16.5%. CMA could detect an additional 11/344 (3.2%) genetic abnormalities compared to the karyotype analysis. Among these cases, 18 involved pathogenic or likely pathogenic CNVs, and 39 involved numerical chromosomal abnormalities (aneuploidies). We identified a total of 11 cases with CNVs that could not be detected by conventional cytogenetic analysis (Table 2).

The distribution of chromosomal abnormalities across different clinical indications is visually summarized in Figure 1. The highest diagnostic yield was observed in the cystic hygroma group, where chromosomal abnormalities were identified in 83% of cases. This was followed by cases with high-risk NIPT results (58%) and those presenting with multiple sonographic findings (26%). Notably, the group with CHD showed a substantial yield of 37%, whereas increased NT and CNS anomalies yielded lower detection rates of 15% and 8.8%, respectively (Table 3).

Among the 41 cases with ultrasound findings, abnormalities were detected in 7.3% of cases, including 2 pathogenic CNVs and 1 aneuploidy. Congenital anomalies were identified in 39 cases, with a 5.3% abnormality rate.

Notably, in the group with multiple sonographic findings (*n* = 38), the abnormality rate increased significantly to 26%, highlighting the cumulative risk when multiple structural anomalies are present.

In the hydrops fetalis group (*n* = 15), chromosomal abnormalities were detected in two cases (14%), both corresponding to aneuploidies. No pathogenic findings were identified in cases with isolated IUGR (*n* = 5) or amniotic fluid abnormalities (*n* = 3).

Among the nineteen cases that underwent testing due to biochemical risk factors, two chromosomal abnormalities were detected (10.5%). In the “other” category, including cases with positive family history (*n* = 18), the abnormality rate was 17%.

In the AMA group (*n* = 21), only one chromosomal abnormality was detected (4.7%), suggesting a relatively lower diagnostic yield when AMA was the sole indication for testing.

These results underscore the clinical value of performing CMA, particularly in pregnancies with multiple or specific sonographic anomalies; they also emphasize the lower likelihood of pathogenic findings in isolated or less specific indications.

## 4. Discussion

Array-based methods, especially SNP-microarrays, are frequently used in prenatal diagnosis. SNP-microarrays can detect >1 Kb microdeletions and microduplications with a higher resolution than karyotyping, and they do not require cell culture. In 2013, the American College of Obstetrics and Gynecology (ACOG) recommended the use of CMA instead of traditional karyotyping for invasive prenatal diagnosis when one or more ultrasound anomalies are detected in the fetus [13]. In this study, we analyzed the results of 344 prenatal SNP-microarray cases to assess the abnormal findings associated with different prenatal diagnosis indications, and we showed that CMA could detect an additional 11/344 (3.2%) genetic abnormalities compared to the karyotype analysis. This rate is consistent with that identified in previous studies [4,5]. The overall abnormal rate in our cohort was 16.5%, pathogenic or likely pathogenic CNVs were detected in 5.2% of cases, and aneuploidy was identified in 11.3% of cases. In a study conducted by Wapner et al., more than 4000 samples from 29 centers were analyzed, and cases that were reported as having normal karyotypes identified using conventional methods were re-evaluated using CMA [3]. As a result, small deletions and duplications (CNVs) were identified in 6% of cases. This study concluded that the use of CMA is beneficial in diagnosing aneuploidies and unbalanced rearrangements, but it may be insufficient for detecting balanced translocations and triploidy. In a review conducted by Callaway et al., CMA was applied to pregnant women whose conventional karyotyping produced normal results [14]. The rate of CNV detection ranged between 0.8% and 5.5%, with an average rate of 2.4%. In these pregnant women, the incidence of abnormal fetal ultrasound findings ranged from 6.0% to 11.1%, with an average incidence of 6.5%. This review also included an analysis of pregnant women with abnormal fetal ultrasound findings; they reported that CNVs were detected in 7% of fetuses with abnormal US. Based on these findings, the authors stated that CMA should be recommended as a first-line test. In a study conducted in Turkey involving 320 patients, the CNV detection rate was reported to be 12.3% [15]. The abnormality rates varied significantly depending on the clinical indications, highlighting the differential diagnostic yield of prenatal microarray analysis across different risk categories.

In our study, among the highest detection rates, cystic hygroma (83%) and high-risk NIPT results (58%) showed the strongest correlation with chromosomal abnormalities. These findings are consistent with previous studies suggesting that cystic hygroma is frequently associated with aneuploidy, particularly Turner syndrome and trisomy 21, 18, or 13 [16]. Similarly, the high diagnostic yield in cases with abnormal NIPT results underscores the efficacy of using NIPT as a screening tool for common chromosomal aneuploidies. A subset of pregnancies with high-risk NIPT results showed no detectable chromosomal abnormalities by karyotyping or CMA. Several mechanisms may explain this discrepancy, including confined placental mosaicism, maternal copy number variations, or technical artifacts inherent to NIPT [16,17]. Additionally, NIPT is primarily designed to screen for a limited number of chromosomal abnormalities and cannot reliably detect balanced rearrangements or low-level mosaicism. Previous studies have reported false-positive NIPT results, particularly in cases with placental–fetal discordance [18].

Cases with multiple indications (26%) and those with major structural anomalies such as CNS abnormalities (8.8%), as well as cases with increased NT (15%), showed a higher abnormality rate. This highlights the importance of performing a detailed fetal ultrasound evaluation in order to guide prenatal genetic testing. The presence of congenital anomalies as a standalone indication yielded a lower diagnostic rate (5.3%). Given that congenital anomalies are known to generally have multifactorial inheritance, this finding is not surprising (Figure 2).

As expected, cases with skeletal anomalies (3.8%) showed a relatively lower rate of abnormal findings. This may reflect the limitations of chromosomal microarrays in detecting single-gene disorders or non-structural genetic etiologies associated with these phenotypes. Similarly, IUGR and anhydramnios/oligohydramnios cases did not yield any abnormal microarray results in our cohort. This may be due to the limited sample size of our cohort, which consists of only 344 patients. However, it should also be considered that such conditions may result from multifactorial inheritance or non-genetic etiologies. But also, the sample sizes in certain subgroups, such as cystic hygroma (*n* = 6) and IUGR (*n* = 5), are relatively small for determining diagnostic yield rates. This inevitably affects the reliability of the subgroup diagnostic yield estimates (e.g., 0% in IUGR). However, these small subgroups reflect the actual distribution of clinical indications referred to our center during the study period. Nevertheless, including these cases is important for providing a comprehensive representation of the real clinical spectrum encountered in high-risk pregnancies evaluated with karyotyping and CMA.

In a subset of cases in our cohort (Table 1), the phenotypic features observed were atypical for the identified chromosomal abnormality. For instance, findings such as pulmonary stenosis, cleft lip and palate, renal pelviectasis, thymic hypoplasia, or D-transposition of the great arteries are not typically associated with Klinefelter syndrome. These atypical manifestations may be coincidental or suggest the presence of a secondary genetic etiology, such as a monogenic disorder or complex chromosomal rearrangement beyond the resolution of CMA [19,20]. While CMA can detect complex rearrangements involving multiple chromosomes and cryptic CNVs [21], in our cohort, no additional pathogenic or likely pathogenic variants were identified in these atypical cases. However, other genetic mechanisms—including non-coding variants, epigenetic alterations, or polygenic effects—remain possible and could be elucidated via further analyses such as exome or whole-genome sequencing.

In our cohort, soft ultrasound markers displayed a relatively high diagnostic yield for chromosomal abnormalities, as seen in Figure 1. This elevated performance can be attributed to findings—such as increased NT, echogenic bowel (EB), and other subtle markers—being more strongly associated with chromosomal changes. For instance, a recent large meta-analysis found that CMA detected pathogenic CNVs in 3.9% of cases with isolated soft markers, notably in fetuses presenting septal defects (SF) [22]. Another study showed that fetuses with increased NT (≥3 mm) had a pCNV detection rate of 9.1%, of which around 6% were submicroscopic changes undetectable via standard karyotyping [23]. These findings underscore the added diagnostic value of CMA in pregnancies that exhibit these markers, even when major structural anomalies are absent.

VOUS remains an inherent challenge in CMA-based prenatal diagnostics. Although our reporting policy, by ACMG recommendations, was limited to pathogenic and likely pathogenic variants, the detection of VUS can lead to considerable clinical dilemmas, including difficulties in genetic counseling, parental anxiety, and challenges in decision-making under uncertainty. In addition, CMA carries ethical considerations, such as the incidental detection of findings unrelated to the initial indication for testing, the identification of variants associated with late-onset disorders, and susceptibility loci with incomplete penetrance. These findings may have significant implications for reproductive choices and postnatal management, underscoring the importance of pre- and post-test counseling to ensure informed decision-making [24].

Our findings emphasize the importance of selecting appropriate prenatal genetic testing strategies based on clinical indications. In cases with USG findings in addition to prenatal diagnosis indications, such as biochemical risk or AMA, CMA should be planned simultaneously with prenatal diagnosis. While microarray analysis provides higher-resolution chromosomal anomaly detection and a more precise phenotype expectation, complementary approaches such as whole-exome sequencing (WES) or targeted gene panels may be required in cases with suspected monogenic disorders. Additionally, the identification of pathogenic and likely pathogenic CNVs in certain cases highlights the necessity of accurate genetic counseling to discuss potential implications for fetal prognosis and familial recurrence risks.

As far as we know, our study is the largest and most comprehensive conducted in Türkiye. Future studies with larger cohorts, along with the integration of CMA methods into the prenatal diagnosis process, will be crucial for further refining the prenatal diagnostic approach. Our results contribute to the growing body of evidence supporting the role played by prenatal microarray analysis, particularly in high-risk pregnancies with structural anomalies or positive NIPT findings.

## 5. Conclusions

Our study highlights the diagnostic value of performing prenatal CMA in a cohort of 344 cases with various clinical indications. The overall abnormality detection rate was 16.5%, with significant variations across different prenatal indications. The highest diagnostic yields were observed in cases with cystic hygroma and high-risk NIPT results, confirming the strong association between these findings and chromosomal abnormalities. In contrast, indications such as skeletal anomalies, isolated congenital anomalies, and AMA showed lower detection rates, suggesting that additional genetic testing approaches, such as WES or targeted gene panels, may be necessary in selected cases.

Our findings reinforce the importance of integrating prenatal US, biochemical screening, and NIPT results into the decision-making process for genetic testing. The identification of pathogenic and likely pathogenic CNVs in certain cases highlights the necessity of patients undergoing comprehensive genetic counseling to discuss clinical implications and recurrence risks.

As prenatal genetic testing continues to advance, we expect that future research involving larger cohorts and advanced genomic technologies will play a crucial role in refining diagnostic strategies. Expanding the use of genome-wide sequencing approaches may enhance our ability to detect underlying genetic etiologies in fetuses with unexplained structural anomalies. Overall, our results contribute to the growing body of evidence supporting the role played by prenatal microarray analysis in high-risk pregnancies and emphasize the need for taking a personalized, multidisciplinary approach in prenatal diagnosis.

## Figures and Tables

**Figure 1 diagnostics-15-02129-f001:**
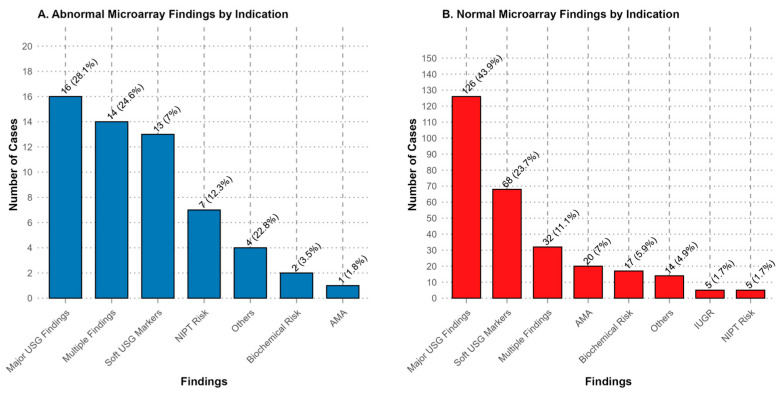
Patient distribution of abnormal and normal result groups in terms of indications. (**A**) Abnormal microarray findings were most frequently detected in cases with major ultrasonography (USG) findings (28.1%), multiple findings (24.6%), and soft USG markers (22.8%). (**B**) Normal microarray findings were most commonly observed in cases with major USG findings (43.9%), soft USG markers (23.7%), and multiple findings (11.1%). Percentages indicate the proportion of cases within the respective category.

**Figure 2 diagnostics-15-02129-f002:**
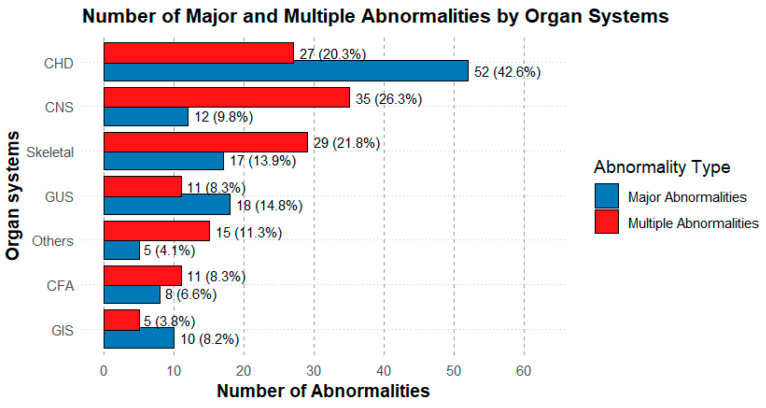
Distribution of the number of major abnormalities by organ system. The graph shows the number of major abnormalities for each organ system, classified as Multiple Abnormality Group (red) and Major Abnormality Group (blue). The most frequent abnormalities were the CHD, followed by the CNS and musculoskeletal system abnormalities. Percentages represent the proportion within each group’s total.

**Table 1 diagnostics-15-02129-t001:** Abnormal results of all of the patients.

Sample	Results (Hg19)	Week	USG	Maternal Age	Group
AS	16p13.11(14975292_16295863)x1	23 + 4	Enlarged ventricle	30	USG findings
AS	16p12.2p11.2(21575087_29319922)x1	26	Enlarged ventricle	35	USG findings
AS	Trisomy 21	21 + 2	Hepatic calcification, echogenic cardiac focus	28	USG findings
AS	15q11.2(22766739_23226254)x1	22	Ambiguous genitals, hydronephrosis	20	Congenital anomaly
AS	Trisomy 21	23 + 2	Renal pyelectasis	33	Congenital anomaly
AS	Trisomy 13	24 + 3	Ventriculomegaly, renal pyelectasis, hypospadias, coarctation of the aorta	38	Multiple findings
AS	Trisomy 18	30	Anal atresia, polyhydramnios, IUGR, single umbilical artery	24	Multiple findings
AS	Trisomy 21	18 + 3	Renal pelviectasis, AVSD ^a^	36	Multiple findings
AS	8p23.3p23.1(170692_12009597)x3 9p24.3p11.2(10201_44888946)x3, 9q13q22.33(68158106_101087286)x3	16 + 6	Cleft palate, CHD	35	Multiple findings
AS	Klinefelter Syndrome	22	Pulmonary stenosis, cleft lip and palate, renal pelviectasis, thymus hypoplasia	26	Multiple findings
AS	Trisomy 18	28	Clenched hand, VSD	26	Multiple findings
AS	Trisomy 18	22	IUGR, clench hand, mandibular hypoplasia, VSD, horseshoe kidney	35	Multiple findings
AS	Trisomy 21	17	Duodenal atresia, NT:6 mm	39	Multiple findings
AS	Trisomy 13	23	Inferior vermis hypoplasia, polyhydramnios, mesocardia, TGA ^b^	37	Multiple findings
Chord sample	Trisomy 13	24	Cleft lip/palate, hyperecogenic bowel, hypoplastic left heart, aortic coarctation, holoprosencephaly	24	Multiple findings
AS	22q11.21(18877787_21461607)x1, Di George	28	Truncus arteriosus, hypoplastic thymus, VSD	35	CHD
AS	Trisomy 21	21 + 4	Hypoplastic nasal bone, AVSD ^a^	33	CHD
AS	14q32.2q32.33(99718925_107289511)x1	32	Craniosynostosis, hypoplastic left heart, aortic hypoplasia, doubled collecting system of the left kidney	25	CHD
AS	Klinefelter Syndrome	16 + 3	TGA ^b^	38	CHD
AS	Turner Syndrome	25	Aortic hypoplasia	21	CHD
AS	4p16.3p11(84414_49620838)x3, 13q11q12.11(19020095_21578150)x1	23 + 2	Pulmonary hypoplasia, VSD, Fallot tetralogy, overriding aorta, clenched hand	23	CHD
CVS	Turner Syndrome	14 + 1	Hypoplastic left heart	23	CHD
AS	Trisomy 13	21 + 5	AVSD ^a^	23	CHD
AS	Trisomy 21	17 + 2	VSD, echogenic liver focus	37	CHD
AS	11q23.3q25(119110984_134946504)x1, 11q23.3(118545797_119103406)x3	22 + 4	Hypoplastic left heart	31	CHD
AS	Xq27.2q28(140856453_155234707)x1, 4q28.3q35.2(134134331_190484505)x3	23	VSD, truncus arteriosus, left-sided gall bladder	33	CHD
AS	Trisomy 18	22	IUGR, Perimembranous VSD	35	CHD
AS	13q21.33q33.2(73157290_105760332)x1	30	Vermian hypoplasia, Pes equinovarus	25	CNS anomaly
AS	16p11.2(29323692_30364805)x3	22 + 5	hydrocephaly, lemon sign, cerebellar hypoplasia, left multicyclic dysplastic kidney, sacral meningomyelocele.	37	CNS anomaly
AS	Trisomy 21	16 + 5	Alobar holoprosencephaly	37	CNS anomaly
CVS	Trisomy 18	12	NT:7 mm	43	Increased NT
CVS	Trisomy 21	12 + 5	NT:5, cystic hygroma	32	Increased NT
AS	10p11.1(38784659_39150257)x1, 10q11.22q11.23(49262918_51832748)x1	16	NT 2.6	36	Increased NT
AS	Trisomy 21	13 + 4	NT 5, diffuse edema, echogenic cardiac focus	38	Increased NT
CVS	4q31.3q35.2(155190509_191044208)x3	13	NT:4 mm	39	Increased NT
AS	Trisomy 13	18	Polydactyly of the right foot, hyperecogenic heart	37	Skeletal anomaly
AS	Xp22.31(6453470_8126718)x0	15 + 4	N	23	Biochemical risk
AS	4q22.2q22.3(94006191_97808388)x1	17	N	34	Biochemical risk
AS	15q11.2(22766739_23226254)x1	22	N	20	Other
AS	47,XYY	20 + 4	CSP ^c^	38	Other
CVS	6q14.3q22.31(85761559_120871846)x1	NA	NA	25	Other
AS	Mosaic UPD of chromosome 3	20	CSP ^c^	41	Other
AS	Trisomy 18	17	Megacystic, clenched hand, hydrops, club foot, VSD	40	Hydrops
AS	Trisomy 21	28	Hydrops, polyhydramnios	34	Hydrops
AS	Yp11.31p11.2(2657176_10057648)x2,Yq11.1q11.221(13133499_19567718)x2,Yq11.222q11.223(20804835_24522333)x0	20	N	24	NIPT risk
AS	Trisomy 21	19 + 2	Fallot tetralogy	35	NIPT risk
CVS	Trisomy 21	13 + 5	NIPT Tr.21 risk	24	NIPT risk
AS	16q11.2q23.1(46501717_75493481)x3	14	N	24	NIPT risk
AS	Trisomy 21	17	NT:3.4 MM	17	NIPT risk

^a^ Atrioventricular septal defect; ^b^ transposition of great arteries; ^c^ cavum septi pellucidi.

**Table 2 diagnostics-15-02129-t002:** Abnormal result ratios of prenatal microarrays where conventional karyotyping was normal.

Results (Hg19)	Size	Detected by Karyotyping	Week	USG
16p13.11(14975292_16295863)x1	1.32 Mb	No	23 + 4	Enlarged ventricle
16p12.2p11.2(21575087_29319922)x1	7.74 Mb	Yes	26	Enlarged ventricle
15q11.2(22766739_23226254)x1	460 Kb	No	22	Ambiguous genitals, hydronephrosis
8p23.3p23.1(170692_12009597)x3 9p24.3p11.2(10201_44888946)x3 9q13q22.33(68158106_101087286)x3	11.8 Mb 44.8 Mb 33 Mb	Yes	16 + 6	Cleft palate, CHD
22q11.21(18877787_21461607)x1	2.583 Kb	No	28	Truncus Arteriosus, hypoplastic thymus, VSD
14q32.2q32.33(99718925_107289511)x1	7.5 Mb	Yes	32	Craniosynostosis, hypoplastic left heart, aortic hypoplasia, doubled collecting system of the left kidney
4p16.3p11(84414_49620838)x3, 13q11q12.11(19020095_21578150)x1	49.5 Mb 2.6 Mb	Yes No	23 + 2	Pulmonary hypoplasia, VSD, Fallot tetralogy, overriding aorta, clenched hand
11q23.3q25(119110984_134946504)x1 11q23.3(118545797_119103406)x3	15.8 Mb 558 Kb	Yes No	22 + 4	Hypoplastic left heart
Xq27.2q28(140856453_155234707)x1 4q28.3q35.2(134134331_190484505)x3	14.2 Mb 56.3 Mb	Yes	23	VSD, truncus arteriosus, left-sided gall bladder
13q21.33q33.2(73157290_105760332)x1	33 Mb	Yes	30	Vermian hypoplasia, Pes equinovarus
16p11.2(29323692_30364805)x3	1.04 Mb	No	22 + 5	Hydrocephaly, lemon sign, cerebellar hypoplasia, left multicyclic dysplastic kidney, Sacral meningomyelocele.
10p11.1(38784659_39150257)x1, 10q11.22q11.23(49262918_51832748)x1	366 Kb 2.6 Mb	No	16	NT 2.6
4q31.3q35.2(155190509_191044208)x3	36 Mb	Yes	13	NT:4mm
Xp22.31(6453470_8126718)x0	1.7 Mb	No	15 + 4	N
4q22.2q22.3(94006191_97808388)x1	3.8 Mb	No	17	N
15q11.2(22766739_23226254)x1	460 Kb	No	22	N
6q14.3q22.31(85761559_120871846)x1	35.1 Mb	Yes	NA	NA
Mosaic UPD of whole chromosome 3		No	20	CSP
Yp11.31p11.2(2657176_10057648)x2, Yq11.1q11.221(13133499_19567718)x2, Yq11.222q11.223(20804835_24522333)x0	7.4 Mb 6.4 Mb 3.7 Mb	Yes	20	N
16q11.2q23.1(46501717_75493481)x3	29 Mb	Yes	14	N

**Table 3 diagnostics-15-02129-t003:** Detection rates of chromosomal abnormalities according to clinical indication.

Indications	N	Abnormal	P/LP ^a^ CNV	Aneuploidy	Abnormal Rate (%)
USG findings	41	3	2	1	0.073
Congenital anomaly	39	2	1	1	0.053
Multiple indications	38	10	1	9	0.26
CHD	35	13	5	8	0.37
CNS anomaly	34	3	2	1	0.088
Increased NT	33	5	2	3	0.15
Skeletal anomaly	26	1	-	1	0.038
Biochemical risk	19	2	1	1	0.105
Other (family history)	18	3	2	1	0.17
Hydrops	15	2	-	2	0.14
NIPT risk	12	7	2	5	0.58
AMA	21	1	-	1	0.047
Cystic hygroma	6	5	-	5	0.83
IUGR	5	-	-	-	0
Anhydramnios/oligohydramnios	3	-	-	-	0
Total	344	57	18	39	0.165

^a^ P: Pathogenic, LP: Likely Pathogenic.

## Data Availability

The data used and analyzed during this study are available from the corresponding author upon reasonable request.

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
