# Peer review of "The Diagnostic Utility of Prenatal Microarray in High-Risk Pregnancies: A Single-Center Experience in Enhancing Reproductive Care and Risk Stratification"

_diagnostics, 2025, doi:10.3390/diagnostics15172129_

Round 1
Reviewer 1 Report
Comments and Suggestions for Authors
In this work, Bakır et al. report the results of 344 fetuses subjected to prenatal conventional karyotyping and SNP microarray or CMA analysis under various prenatal diagnosis indications. The observed overall diagnostic yield by employing both methodologies was 16.5%, involving the identification of 3.2% (11/344) of cases bearing pathogenic/likely pathogenic copy number variants (P/LP CNVs) detected only by CMA. These chromosomal abnormalities were highly prevalent in fetuses with cystic hygroma (83%) or in those derived from gestations with high-risk NIPT results (58%).
Their results agree with several previous figures reported in the prenatal diagnostic setting. However, the manuscript must be improved in the following aspects:
- Please consider reviewing the manuscript by a native English language editor. Along text, there are several typos (i.e., “fetüs”, lane 180, “Aneuploidi”,"Megacystit”, “hipoplasia”, among others in tables 1 and 2).
- Abbreviations in the text must be used accordingly to the provided “Abbreviations list” section. Please standardize their use along text (i.e. “AMA”, “NIPT”, “CHD”, “NT”, and so on).
- Lane 73: Please check the accuracy of the sentence “coronary heart disease” vs. “congenital heart disease”.
- In the Materials and Methods section, please describe the analyzed number of metaphases and the G-banding resolution used for karyotyping studies.
- In antecedents, please clarify that other similar studies have been carried out in Turkey to avoid further confusion in the discussion section. I.e. “Additionally, no studies from Turkey have been found that include a cohort of this size.”, but in Discussion section, the authors discuss a very similar study (in methodology and number of patients analyzed): “In a study conducted in Turkey involving 320 patients, the CNV detection rate was 197 reported to be 12.3% [15].”
- Seems to be very interesting to discuss those cases derived from pregnancies with a high-risk NIPT result, but without any detectable chromosomal abnormality; thus, what happened with the remaining 42% of “chromosomally normal fetuses?. Please discuss or describe briefly.
- Please describe if in any of your studied cases, a trio analysis was warranted to classify any CNV of uncertain significance.
- In the conclusion section, please verify the accuracy of the following sentences:
“Our study highlights the diagnostic value of prenatal chromosomal microarray analysis in a cohort of 344 cases with various clinical indications. The overall abnormality detection rate was 16.5%, with significant variations across different prenatal indications.” Please note that the cited overall diagnostic yield includes the karyotype evaluation. Thus, to avoid confusion or misunderstanding of the main contributions of this work, the authors must separate the diagnostic yield of either karyotype or CMA, and the combined diagnostic yield, as CMA contributes to a lesser extent (<4%) to the prenatal diagnosis than karyotype.
- Some of the reported fetuses (Table 1) have atypical phenotypes that do not correlate with the identified chromosomal abnormality (i.e., Pulmonary stenosis, cleft lip and palate, renal pelviectasis, thymus hypoplasia or D-TGA in Klinefelter síndrome). Can you discuss briefly these atypical cases?, CMA analysis did not reveal the eventual presence of a second complex chromosomal rearrangement, as has been previously noted in literature (i.e., PMID:31388356; 37834089).
- Please check the description of Table 4. The title of this table must be more descriptive to the reader.
- I highly recommend that the clinical and genotypic data in Tables 1 and 4 must be unified in a single table. This new one could be most informative and easy to understand.
- Please discuss briefly the relatively high diagnostic yield correlation between the presence of “soft USG findings” and the presence of chromosomal abnormalities (Figure 1).
The manuscript needs a deep review by an English native language editor.
Author Response
Response to Reviewer 1 Comments
- Summary
We would like to sincerely thank Reviewer 1 for the time and effort dedicated to reviewing our manuscript and for providing constructive and insightful comments. Your feedback has been invaluable in improving the clarity, accuracy, and overall quality of our work. We have carefully considered each point raised and have revised the manuscript accordingly. Below, we provide a point-by-point response, outlining the changes made and clarifying our approach where necessary. All modifications are highlighted in the revised manuscript for ease of reference.
- Questions for General Evaluation
|
Question |
Reviewer’s Evaluation |
Response and Revisions |
|
Does the introduction provide sufficient background and include all relevant references? |
Yes |
Introduction reviewed; sentence on lack of Turkish studies removed for accuracy. |
|
Are all the cited references relevant to the research? |
Yes |
References reviewed and confirmed. |
|
Is the research design appropriate? |
Yes |
No changes required. |
|
Are the methods adequately described? |
Can be improved |
Added metaphase count and G-banding resolution to Methods. |
|
Are the results clearly presented? |
Yes |
Minor clarifications added for diagnostic yield. |
|
Are the conclusions supported by the results? |
Can be improved |
Separated diagnostic yields for karyotype, CMA, and combined. |
- Point-by-Point Response to Comments and Suggestions for Authors
Comment 1: Please consider reviewing the manuscript by a native English language editor.
Response: Language editing was provided by MDPI Author Services. The certificate has been attached.
Comment 2: Abbreviations in the text must be used accordingly to the provided “Abbreviations list” section.
Response: Abbreviations were reviewed and standardized. Repeated full terms were replaced with abbreviations. FISH and AMA were added to the list of abbreviations.
Comment 3: Lane 73: Please check “coronary heart disease” vs. “congenital heart disease.”
Response: Relevant terms have been corrected to “congenital heart disease.”
Comment 4: Please describe the analyzed number of metaphases and the G-banding resolution used.
Response: Added to Materials and Methods: “The metaphases obtained after culture were analyzed at the 450–550 GTG band level using the CytoVision® platform.”
Comment 5: Clarify other similar studies in Turkey to avoid confusion.
Response: The sentence “Additionally, we found no studies from Turkey that include a cohort of this size.” has been removed from the Introduction.
Comment 6: Discuss high-risk NIPT cases without detectable abnormalities.
Response: A discussion of false-positive NIPT results, with references, has been added.
Comment 7: Describe if trio analysis was performed for CNVs of uncertain significance.
Response: Added to Materials and Methods: “Variant interpretation was performed, and only pathogenic and likely pathogenic variants were reported. For patients in whom a VOUS was detected, parental testing was not planned in order to avoid prolonging the reporting process.”
Comment 8: Verify conclusion section accuracy regarding diagnostic yield.
Response: Added a statement clarifying that 16.5% represents the combined yield of karyotype and CMA. Included CMA’s additional contribution percentage in the Results.
Comment 9: Discuss atypical phenotypes in relation to chromosomal findings.
Response: A paragraph discussing atypical findings, with references, has been added to the Discussion.
Comment 10: Revise Table 4 description to be more descriptive.
Response: Table 4 removed from the main text (as per another reviewer’s suggestion) and submitted as supplementary material.
Comment 11: Merge Tables 1 and 4 for clarity.
Response: Table 4 removed from the main text (as per another reviewer’s suggestion) and submitted as supplementary material.
Comment 12: Discuss the high diagnostic yield correlation between soft USG findings and chromosomal abnormalities.
Response: A paragraph discussing this correlation, with references, has been added to the Discussion.
- Response to Comments on the Quality of the English Language
The manuscript underwent full language editing by MDPI Author Services; the editing certificate is attached.
- Additional Clarifications
No further clarifications beyond the above responses are necessary at this stage.
Reviewer 2 Report
Comments and Suggestions for Authors
The study by Bakır et al. presents results of testing 344 prenatal samples (rather large cohort) using both conventional karyotyping and SNP-based chromosomal microarray analysis (CMA). The relevance of this study lies in its direct contribution to optimizing prenatal genetic diagnostics for high-risk pregnancies, particularly in a previously underrepresented population such as Türkiye.
The methodology of this study is robust - using standardized GSA-Cyto platform and NxClinical software with established databases. Authors provided detailed clinical indication breakdowns with practical diagnostic yield percentages and included both statistical analysis (SPSS/R) and visual data presentation (tables/figures). The results clearly demonstrated CMA's added value over karyotyping (11 undetectable CNVs). Authors presented comprehensive data including detailed tables stratify abnormalities by indication (Table 3), organ system (Table 4), and CMA-vs-karyotype findings (Table 2). The study identifies high-yield indications: cystic hygroma (83%) and high-risk non-invasive prenatal testing (NIPT) (58%) benefit most from CMA. Also, the results highlighted low-yield scenarios (e.g., isolated intrauterine growth restriction, advanced maternal age alone), guiding cost-effective test selection. Authors discussed practical implications for test selection based on indication. The key takeaway is that CMA is superior to karyotyping for detecting submicroscopic CNVs, particularly with ultrasound anomalies/NIPT risks.
I don't have many comments or questions on the substance of the manuscript. Rather, it requires some refinement in terms of presentation.
The manuscript is pretty well written, but does not follow the template adopted for MDPI journals. Tables and figures should be integrated into the text. It might make sense to put the large Table 4 in the supplementary material and make the most important extracts from it in the text. The manuscript is replete with abbreviations, some of which are deciphered, some of which are not, such as ARSA (Table 4, Line 330). The authors should pay careful attention to this aspect, as it is difficult for readers to keep track of so many abbreviations. The list of references should also be arranged according to manuscript rules. Table 1 has missing entries (e.g., results for craniosynostosis case).
Materials and methods. Labs maternal contamination was checked (lines 120-121) but didn’t report contamination rates.
Figures. Figure legends (???) (Page 16) lack axis labels, making graphs hard to interpret. Figure 1: make figure a) and b), add in the caption what is now written at the top. The quantities (and percentages) look a bit messy (and for Figure 2 too).
Discussion. Authors should highlight the limitations of their study. First, key cohorts (e.g., cystic hygroma, n=6; IUGR, n=5) are small, reducing reliability of subgroup analyses (e.g., 0% yield in IUGR). The section fails to address variants of uncertain significance (VUS) or ethical dilemmas, a known limitation of CMA. No cost-effectiveness analysis despite advocating for CMA as first-tier testing.
Author Response
Response to Reviewer 2 Comments
- Summary
We would like to sincerely thank Reviewer 2 for the careful review of our manuscript and for the constructive feedback provided. Your comments have helped us refine the presentation and clarity of the study. We have addressed each point in detail, revising the manuscript accordingly. All modifications are highlighted in the revised manuscript for ease of reference.
- Questions for General Evaluation
|
Question |
Reviewer’s Evaluation |
Response and Revisions |
|
Does the introduction provide sufficient background and include all relevant references? |
Yes |
No changes required. |
|
Is the research design appropriate? |
Yes |
No changes required. |
|
Are the methods adequately described? |
Yes |
Added clarifications on contamination policy. |
|
Are the results clearly presented? |
Can be improved |
Figures revised, abbreviations standardized, table formatting improved. |
|
Are the conclusions supported by the results? |
Yes |
No changes required. |
|
Are all figures and tables clear and well-presented? |
Must be improved |
Figures reformatted, legends expanded, Table 4 moved to supplementary material. |
- Point-by-Point Response to Comments and Suggestions for Authors
Comment 1: The manuscript is pretty well written, but does not follow the template adopted for MDPI journals. Tables and figures should be integrated into the text. It might make sense to put the large Table 4 in the supplementary material and make the most important extracts from it in the text.
Response: Tables and figures have been integrated into the main text. Table 4 has been removed from the main text and will be submitted separately as supplementary material.
Comment 2: The manuscript is replete with abbreviations, some of which are deciphered, some of which are not, such as ARSA (Table 4, Line 330). The authors should pay careful attention to this aspect, as it is difficult for readers to keep track of so many abbreviations.
Response: Abbreviations were reviewed and standardized. Repeated full terms in the text were replaced with abbreviations. FISH and AMA were added to the list of abbreviations. Table 4 has been removed from the main text and will be submitted separately as supplementary material.
Comment 3: The list of references should also be arranged according to manuscript rules.
Response: The official MDPI EndNote style was downloaded and used to update the reference list by MDPI guidelines.
Comment 4: Table 1 has missing entries (e.g., results for craniosynostosis case).
Response: All rows and columns were checked, and no missing entries were identified.
Comment 5 (Materials and Methods): Lab's maternal contamination was checked (lines 120–121), but didn’t report contamination rates.
Response: Contaminated materials were not included in the study. A new sample was requested from patients in whom contamination was detected, and repeat testing was planned for those who agreed to provide a new sample. No statistics regarding the contamination rate were recorded by us.
Comment 6 (Figures): Figure legends (Page 16) lack axis labels, making graphs hard to interpret.
Response: Figure legends have been expanded, and axis labels have been added.
Comment 7: Figure 1: make figures a) and b), add in the caption what is now written at the top. The quantities (and percentages) look a bit messy (and for Figure 2, too).
Response: Figure 1 has been separated into plots A and B, and the caption has been expanded. Figure 2 has been recreated, and its captions have been revised for clarity.
Comment 8 (Discussion): Authors should highlight the limitations of their study. First, key cohorts (e.g., cystic hygroma, n=6; IUGR, n=5) are small, reducing reliability of subgroup analyses (e.g., 0% yield in IUGR).
Response: A statement has been added to the Discussion acknowledging that the total sample size and the small size of some subcohorts limit the reliability of subgroup diagnostic yield rates. We emphasized that these small subgroups reflect the actual distribution of clinical indications referred to our center during the study period, and including them provides a comprehensive presentation of the real clinical spectrum encountered in high-risk pregnancies evaluated with karyotyping and CMA.
Comment 9: The section fails to address variants of uncertain significance (VUS) or ethical dilemmas, a known limitation of CMA.
Response: Although VOUS findings were present in our cohort, our reporting policy, by ACMG guidelines, was limited to pathogenic and likely pathogenic variants; thus, these findings were not the primary focus of our analysis. However, we acknowledge that VOUS results and their interpretation can pose significant challenges in counseling, cause parental anxiety, and necessitate decision-making under uncertainty. CMA also has ethical implications, such as the detection of incidental findings unrelated to the referral indication, variants associated with late-onset diseases, or incompletely penetrant susceptibility loci. To highlight the clinical relevance of these issues, we have added a brief discussion of these aspects to the revised manuscript.
Comment 10: No cost-effectiveness analysis despite advocating for CMA as first-tier testing.
Response: Although CMA is increasingly recommended as a first-tier test for high-risk pregnancies, no formal cost-effectiveness analysis was performed in our study, primarily because our retrospective design focused on diagnostic yield and the clinical spectrum of abnormal findings, which was outside the scope of such an analysis.
- Response to Comments on the Quality of the English Language
The manuscript underwent full language editing by MDPI Author Services; the editing certificate is attached.
- Additional Clarifications
No further clarifications beyond the above responses are necessary.
Round 2
Reviewer 1 Report
Comments and Suggestions for Authors
The manuscript was greatly improved regarding English grammar/style, organization, and clarity of description of results and their discussion.
All comments were correctly addressed, just please note that use of some abbreviations are not consistently used throughout the text (i.e. CMA, WES, SF, and some others). Please attend to this point.
Author Response
Comment:The manuscript was greatly improved regarding English grammar/style, organization, and clarity of description of results and their discussion. All comments were correctly addressed, just please note that use of some abbreviations are not consistently used throughout the text (i.e., CMA, WES, SF, and some others). Please attend to this point.
Response: We appreciate the reviewer's valuable remark. In response, we have carefully reviewed the entire manuscript, and all abbreviations (including CMA, WES, SF, and others) have been standardized for consistent usage throughout the text, tables, and figures. Additionally, abbreviations for terms appearing only once have been removed to avoid redundancy. All modifications have been highlighted in the revised manuscript.